# Hyperspectral Unmixing for Raman Spectroscopy
## via Physics-Constrained Autoencoders

**Dimitar Georgiev** [1 2]  **Álvaro Fernández-Galiana** [2 *]  **Simon Vilms Pedersen** [2 *]  **Georgios Papadopoulos** [1 2]
**Ruoxiao Xie** [2]  **Molly M. Stevens** [2 3]  **Mauricio Barahona** [4]

## Abstract

Raman spectroscopy is widely used across life and material sciences to characterize the chemical composition of samples in a nondestructive, label-free manner. Many applications entail the unmixing of signals from mixtures of molecular species to identify the individual components present and their proportions, yet conventional methods for chemometrics often struggle with complex mixture scenarios encountered in practice. Here, we develop autoencoder neural network models for hyperspectral unmixing of Raman spectroscopy data, which we systematically validate using synthetic and experimental benchmark datasets we created in-house. Our results demonstrate that autoencoders provide improved accuracy, robustness and efficiency compared to standard unmixing methods. We also showcase the applicability of our approach to complex biological settings by showing improved biochemical characterization of volumetric Raman imaging data from a human leukemia monocytic cell line.

## 1. Introduction

Raman spectroscopy (RS) is a powerful optical modality that facilitates the identification, characterization and quantification of the molecular composition of chemical and biological specimens, offering in-depth insights into their structure and functionality (Movasaghi et al., 2007; Talari et al., 2015; Butler et al., 2016; McCreery, 2005; Smith & Dent, 2019). RS interrogates the vibrational modes of molecules through the analysis of inelastic scattering of monochromatic light from matter, thereby enabling the nondestructive, label-free fingerprinting of chemical species (Koningstein, 2012; Szymanski, 2012; Colthup, 2012; Jones et al., 2019; Bocklitz et al., 2016). As a result, RS has become an important analytical tool in a myriad of scientific domains, from chemistry (Schlücker, 2014; Dodo et al., 2022), biology (Pezzotti, 2021; Smith et al., 2016; Shipp et al., 2017; Cialla-May et al., 2017), and medicine (Kong et al., 2015; Ember et al., 2017; Pence & Mahadevan-Jansen, 2016; Balan et al., 2019; Auner et al., 2018; Tanwar et al., 2021), to materials science (Fernández-Galiana et al., 2023; Kumar, 2012; Weber & Merlin, 2013), pharmacology (Wang et al., 2018; Paudel et al., 2015), environmental science (Halvorson & Vikesland, 2010; Ong et al., 2020; Terry et al., 2022), food quality control (Li & Church, 2014; Pang et al., 2016), and even forensics (Chalmers et al., 2012; Khandasammy et al., 2018; Izake, 2010).

Despite the wealth of information RS affords, the analysis and interpretation of experimental RS data remains a major challenge (Ryabchykov et al., 2018; Guo et al., 2021; Gautam et al., 2015). Many important applications entail the analysis of complex mixtures of molecular species coexisting and interacting at micro- and nanoscales. Such complexity can hinder the qualitative and quantitative investigation of RS measurements, especially when dealing with the biomolecular diversity of biological samples (Byrne et al., 2016; Gautam et al., 2015).

Hyperspectral unmixing (also known as (hyper)spectral deconvolution or multivariate curve resolution) aims to resolve such mixed signals (Li et al., 2017; Olmos et al., 2017) by identifying the individual components present (*endmember identification*) and/or quantifying their proportions (*abundance estimation*). Popular approaches include N-FINDR (Winter, 1999) and Vertex Component Analysis (VCA) (Nascimento & Dias, 2005) for endmember identification, and Non-negative Least Squares (NNLS) (Law-

---

[*]Equal contribution  [1]Department of Computing & UKRI Centre for Doctoral Training in AI for Healthcare, Imperial College London, London, United Kingdom, SW7 2AZ [2]Department of Materials, Department of Bioengineering & Institute of Biomedical Engineering, Imperial College London, London, United Kingdom, SW7 2AZ [3]Department of Physiology, Anatomy and Genetics, Department of Engineering Science, and Kavli Institute for Nanoscience Discovery, University of Oxford, Oxford, United Kingdom, OX1 3QU [4]Department of Mathematics, Imperial College London, London, United Kingdom, SW7 2AZ. Correspondence to: Molly M. Stevens <molly.stevens@dpag.ox.ac.uk>, Mauricio Barahona <m.barahona@imperial.ac.uk>.

*Accepted at the 1st Machine Learning for Life and Material Sciences Workshop at ICML 2024.* Copyright 2024 by the author(s).

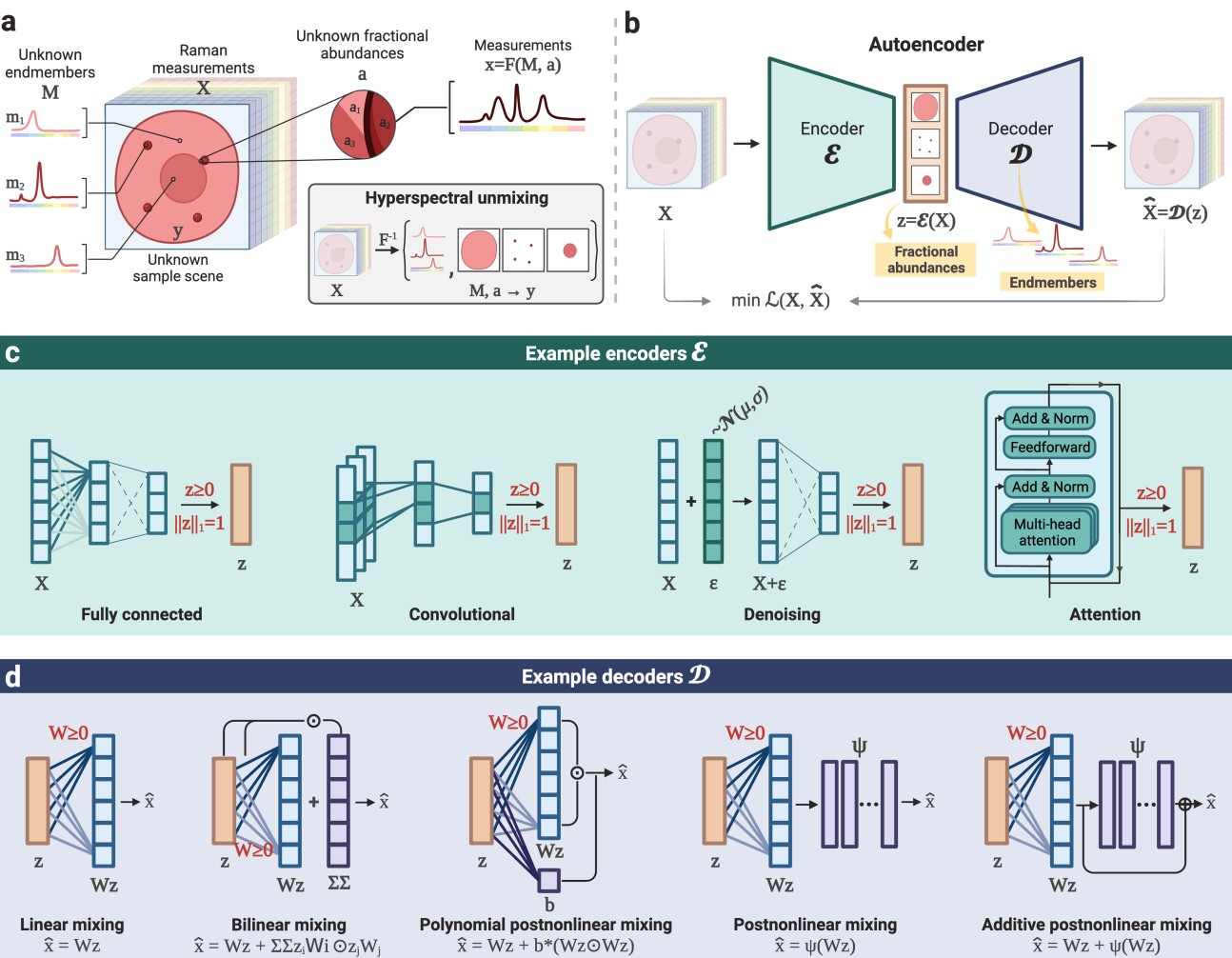

*Figure 1.* Hyperspectral unmixing for Raman spectroscopy using autoencoder neural networks. a, Diagram of the task of hyperspectral unmixing. b, Hyperspectral unmixing as a self-supervised autoencoder learning problem: the decoder learns to derive endmembers and the encoder learns the corresponding fractional abundances. c, Encoders can accommodate different concepts from representation learning, such as convolutional layers and attention, to improve feature extraction and provide more accurate and robust unmixing. d, Decoders can be structured to model different linear and non-linear mixing models.

son & Hanson, 1995) and Fully Constrained Least Squares (FCLS) (Heinz et al., 2001) for abundance estimation (Li et al., 2017; Hedegaard et al., 2011). However, such techniques, which originated in the field of remote sensing (Keshava & Mustard, 2002; Harris, 2006), have limitations for the unmixing of RS data. Specifically, these methods are restricted to linear mixing; lack robustness to data artifacts abundant in RS data (e.g., dark noise, baseline variations, cosmic spikes); rely on additional assumptions (e.g., endmembers present as 'pure pixels' in the data); and are computationally demanding for large datasets (e.g., imaging and volumetric Raman raster scans).

In this work, we introduce an approach for RS hyperspectral unmixing based on autoencoder (AE) neural networks,

which we systematically validate against conventional methods for unmixing using a comprehensive array of synthetic and experimental Raman spectroscopy data.

## 2. Background

**Hyperspectral unmixing in Raman spectroscopy.** Raman spectra can be represented as vectors $\mathbf{x} \in \mathbb{R}_+^b$, whose components correspond to the intensity of inelastically scattered light binned over $b$ wavelength/wavenumber bands. Such measurements can be treated as the result of an underlying mixing of $n$ 'pure' components, defined by their Raman signatures $\mathbf{m}_i \in \mathbb{R}_+^b$, $i = 1, \ldots, n$ (*endmembers*), and their respective proportions $\{\alpha_i\}_{i=1}^n$, $\alpha_i \in \mathbb{R}_+$ (*fractional abundances*). Hyperspectral unmixing is the inverse

problem of recovering the endmembers and fractional abundances from a (collection of) measurement(s) $\mathbf{x}$ (Figure 1a).

Standard methods for unmixing, such as N-FINDR, VCA, NNLS and FCLS, operate under the linear mixing model (LMM), where measurements are assumed to be a linear combination of the endmembers $\mathbf{x} = M\boldsymbol{\alpha} = \sum_{i=1}^{n} \alpha_i \mathbf{m}_i$, where $M = \begin{bmatrix} \mathbf{m}_1 & \mathbf{m}_2 & \cdots & \mathbf{m}_n \end{bmatrix}$ is an $b \times n$ non-negative matrix containing the $n$ endmember signatures, and $\boldsymbol{\alpha} = (\alpha_1, \alpha_2, \cdots, \alpha_n)^{\mathsf{T}}$ is an $n \times 1$ vector storing the corresponding abundances. The abundances $\alpha_i$ are constrained to be non-negative (i.e., the abundance non-negativity constraint (ANC), $\alpha_i \geq 0, \forall i$, and are forced to sum to 1 when corresponding to proportions (i.e., the abundance sum-to-one constraint (ASC), $||\boldsymbol{\alpha}||_1 = 1$ ). Refer to Appendix A for more information about standard techniques for unmixing.

**Autoencoders.** Autoencoders are a family of neural network models consisting of two sub-networks (*encoder* and *decoder*) connected sequentially (Goodfellow et al., 2016). The encoder $\mathcal{E} : \mathbb{R}^b \rightarrow \mathbb{R}^m$, usually $m \ll b$, transforms input data $\mathbf{x}$ to a lower-dimensional latent space representation $\mathbf{z} = \mathcal{E}(\mathbf{x})$, which the decoder $\mathcal{D} : \mathbb{R}^m \rightarrow \mathbb{R}^b$ uses to produce reconstructions $\widehat{\mathbf{x}} = \mathcal{D}(\mathbf{z})$ of the original input. AE models are typically trained in a *self-supervised* manner by minimizing a loss function $\mathcal{L}(\mathbf{x}, \widehat{\mathbf{x}})$ that measures the discrepancy between the input $\mathbf{x}$ and the reconstruction $\widehat{\mathbf{x}}$. During training, the encoder progressively learns a latent representation that captures the most salient features of the input data, whereas the decoder learns how to recover the data from the latent representation. AEs have recently emerged as a framework for hyperspectral unmixing in remote sensing (Palsson et al., 2022; Zhang et al., 2020b; Wang et al., 2022; Bhatt & Joshi, 2020; Chen et al., 2022), yet their utility for RS data remains largely unexplored.

## 3. Raman unmixing autoencoders

The dual functionality of autoencoders can be harnessed to design AE models for hyperspectral unmixing: the latent representations $\mathbf{z} = \mathcal{E}(\mathbf{x})$ can be interpreted as fractional abundances (with respect to an input spectrum $\mathbf{x}$), and the decoder $\mathcal{D}(\cdot)$ acts as a mixing function on these representations by encoding endmember signatures and other interactions (Figure 1b). This setup provides a highly adaptable and versatile framework for unmixing.

**Encoder design.** On the one hand, the learning of physical and biochemical features in the encoder can be enhanced by adopting strategies from representation learning, such as convolutional layers to capture spectral and/or spatial correlations among neighboring bands and/or pixels (Zhang et al., 2018; Palsson et al., 2020; Elkholy et al., 2020), or attention mechanisms to model long-range dependencies (Ghosh

et al., 2022) (Figure 1c). In addition, sparsity, part-based learning and denoising objectives can be adopted to enhance explainability and robustness (Ozkan et al., 2018; Qu & Qi, 2018; Su et al., 2018; 2017; Qu et al., 2017). In this work, we consider four types of encoders encompassing a variety of architectures, from standard dense layers to more contemporary convolutional and attention mechanisms (see Appendix B for more details): 1) an encoder consisting of fully connected layers (*Dense*); 2) an encoder with a 1D convolutional feature extractor block, followed by a fully connected part (*Convolutional*); 3) a transformer-based encoder that uses multi-head attention (*Transformer*) (Vaswani et al., 2017); and 4) a transformer-based encoder with a 1D convolutional feature extractor (*Convolutional Transformer*).

**Decoder design.** On the other hand, the design of the decoder allows for flexible modeling of input data to account for various mixture models, e.g., linear, bilinear and post-nonlinear (Figure 1d) (Chen et al., 2022; Shahid & Schizas, 2021; Zhao et al., 2021), akin to introducing an inductive prior with respect to the mixture model directly via the AE architecture. The two types of decoders we develop are: 1) a decoder for linear unmixing, which consists of a single fully connected layer defined by a $b \times m$ weight matrix $W$, with reconstructions reducing to $\widehat{\mathbf{x}} = W\mathbf{z}$; and 2) a decoder of the same architecture for bilinear unmixing based on the Fan model (Fan et al., 2009), where additional bilinear interaction terms are calculated such that $\widehat{\mathbf{x}} = W\mathbf{z} + \sum_{k=1}^{m} \sum_{\substack{l=1 \\ l \neq k}}^{m} z_k \mathbf{w}_k \odot z_l \mathbf{w}_l$, where $z_k, z_l$ are components of $\mathbf{z}$, and $\mathbf{w}_k, \mathbf{w}_l$ are column vectors of $W$.

**Physics-inspired constraints.** To guide the AE learning and reinforce the physical interpretation of unmixing, we incorporate appropriate physical constraints into the AE architectures, e.g., non-negativity of endmembers and fractional abundances, and sum-to-one abundances. We enforce fractional abundance constraints through the choice of a latent space activation function. We use softmax to enforce both ANC and ASC; or, when interested in ANC only, a 'softly-rectified' hyperbolic tangent function given by $f(x) = \frac{1}{\gamma} \ln(1 + e^{\gamma * tanh(x)})$, $\gamma = 10$, designed to ensure abundances are non-negative (between 0 and 1) but do not necessarily add up to one. To ensure the non-negativity of endmembers, we constrain the weight matrix $W$ in our decoders by clipping negative values to zero.

**Model training and evaluation.** We train our AE models in a self-supervised fashion by minimizing a loss based on spectral angle divergence (SAD) (Kruse et al., 1993) between input and reconstructed spectra. We compare AE performance to conventional unmixing approaches: N-FINDR and VCA as endmember extraction algorithms followed by NNLS or FCLS to derive fractional abundances. When

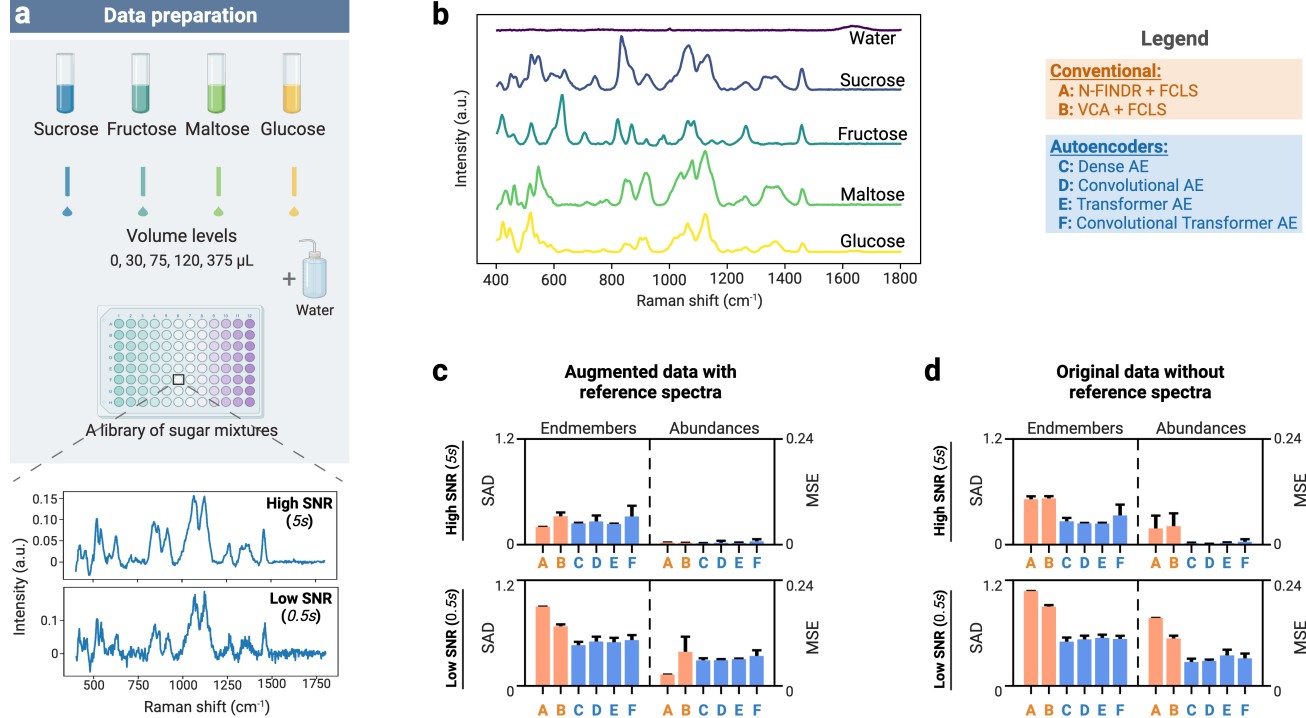

*Figure 2.* Experimental validation on Raman spectroscopy data from sugar solutions. a, Schematic diagram of sugar mixture preparation. Two sets of data are acquired—high and low signal-to-noise ratio (SNR) data, by using integration times of 5 s and 0.5 s, respectively. b, Endmember signatures estimated from reference spectra (high SNR) additionally collected from pure solutions. c-d, Summary of unmixing performance for: an idealized scenario with augmented data including reference spectra (c); and original data without augmentation (d). Confidence intervals are given as one standard deviation around the sample mean ($n = 5$).

ground truth information is available, accuracy is quantified with respect to two measures - mean squared error (MSE) for fractional abundances, and SAD for endmembers. For each evaluation, we first assign derived and ground truth endmembers (and corresponding abundances) via the Hungarian algorithm with SAD as the objective to minimize.

## 4. Experimental validation

To validate the approach, we evaluate the performance of the unmixing AE models we developed on a range of synthetic and experimental RS data.

**Validation on synthetic Raman mixtures.** First, we conduct a systematic validation on synthetic Raman datasets of variable complexity (with *vs* without artifacts, linear *vs* nonlinear mixing, different mixing levels), which we generated in-house using a custom data generator (see Appendix C for further experimental details). We find that our AE models significantly outperform methods like N-FINDR+FCLS and VCA+FCLS across virtually all 11 types of datasets (see Figure 4c-d in Appendix). We also observe that all AE models are faster and less computationally expensive than the two conventional methods (see Figure 5 in Appendix).

**Validation on experimental Raman data from sugar mixtures.** Next, we validate the unmixing performance of AEs on real experimental data. In particular, we conduct benchmark analyses on data from a library of 240 sugar mixtures prepared in-house with four types of sugar (glucose, sucrose, fructose, maltose) at different concentrations (Figure 2a). To consider different signal-to-noise (SNR) conditions, we acquired high SNR (1920 spectra) and low SNR (7680 spectra) measurements using a custom Raman microspectroscopy platform at integration times of 5 s and 0.5 s, respectively. We perform unmixing on these data to identify the content of each mixture, i.e., types of sugar and their concentrations (see Appendix D). The ground truth is defined by the experimental concentrations and the endmember signatures we obtain from reference spectra measured from 5 additional pure solutions (Figure 2b). As with the synthetic data above, we benchmark the performance of our four AE models (linear decoders) against N-FINDR+FCLS and VCA+FCLS.

First, we consider an idealized scenario, purposefully devised to favor conventional methods, whereby endmembers are present in the data. To do this, we augment our data with the additional reference spectra we measured. When such 'pure pixels' are available, we observe that conventional

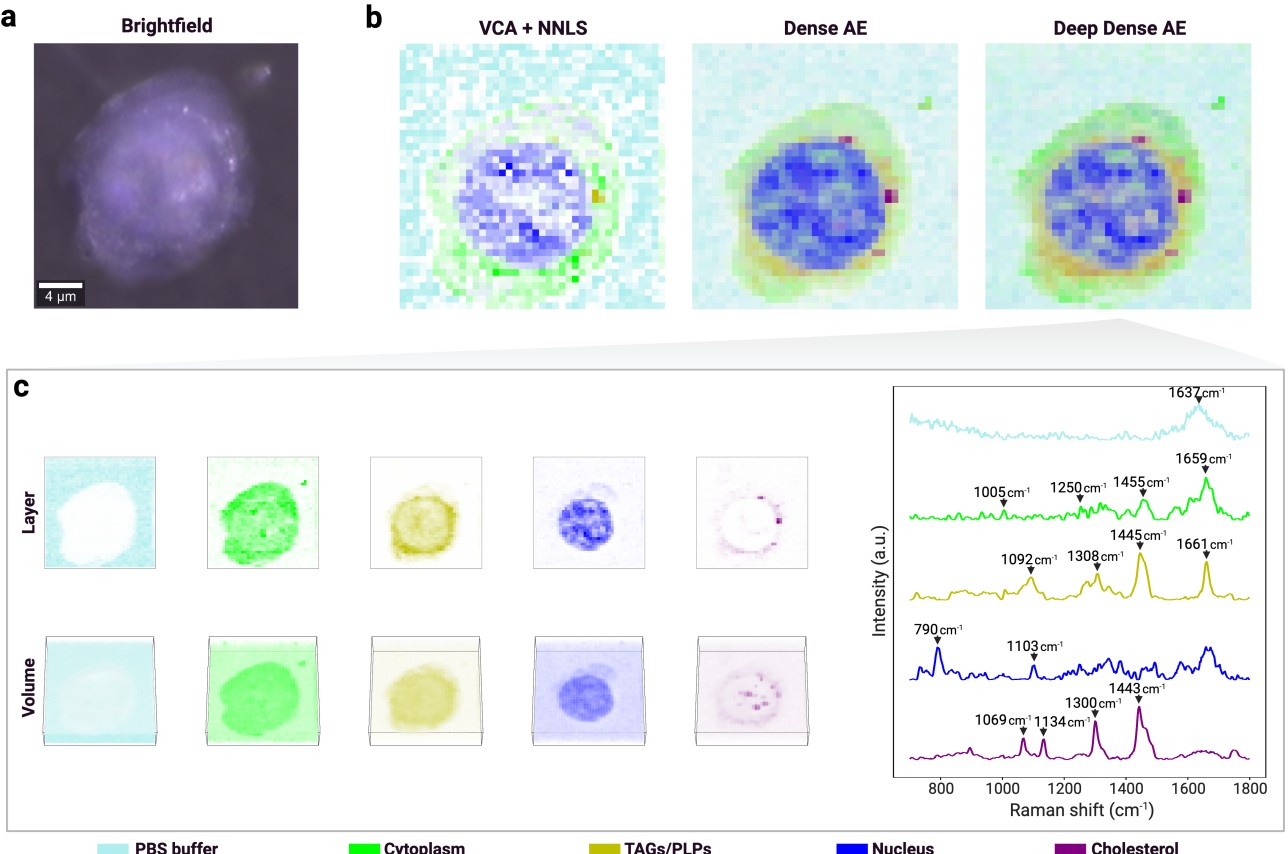

*Figure 3.* Improved volumetric Raman imaging of a THP-1 cell with unmixing autoencoders. a, A brightfield image of the studied THP-1 cell. b, A cross-section reconstruction of the cell (layer $z = 7$) obtained by overlaying the fractional abundances derived by: VCA+NNLS, our *Dense AE*, and our *Deep Dense AE*. c, Results obtained with the *Deep Dense AE* model, displaying the spatial distribution of the individual fractional abundances and the associated endmember signatures. Brightfield and Raman data from Kallepitis et al. (2017).

methods (NFINDR+FCLS, VCA+FCLS) perform comparably to AEs on clean, high SNR data (Figure 2c). Yet, AEs already provide improved performance in low SNR regimes.

In many experimental applications, however, the underlying endmembers are not present in the data and cannot be separately obtained (e.g., target-agnostic applications, or unknown species). To consider this, we analyzed our original data without augmentation. Our results in Figure 2d demonstrate that, in such situations, AEs substantially outperform conventional methods in both low and high SNR settings.

**Real biological application: volumetric Raman imaging of a cell.** Lastly, we use unmixing autoencoders to analyze a low-SNR volumetric RS raster scan of a human leukemia monocytic (THP-1) cell (Figure 3a) (Kallepitis et al., 2017). Using Raman chemometrics, the composition of the cell is probed to study its morphology in a nondestructive, label-free manner. After loading and preprocessing the data using *RamanSPy* (Georgiev et al., 2024), we conduct unmixing with: 1) VCA+NNLS - as in the original paper; 2) *Dense AE*

- our simplest and most computationally efficient AE model; and 3) *Deep Dense AE* - an extension of *Dense AE* with a deeper encoder with five layers. We derive 20 endmembers, which we characterize via peak assignment to identify biochemical species present in the scanned cell, such as deoxyribonucleic acid (DNA), proteins, triglycerides (TAGs), phospholipids (PLPs) and cholesterol esters.

Figure 3b shows the reconstructions of the cell created by overlaying selected fractional abundances derived by each method, revealing the spatial organization of key cellular organelles, including the nucleus, cytoplasm, lipid bodies and membranes. Although direct comparisons are challenging due to the lack of ground truth, we observe that our AE models, especially our *Deep Dense AE*, enable more precise spectral and compositional information (Figure 3c). Notably, unlike the original VCA+NNLS approach, our AEs detect cholesterol, an important functional and structural component in cells (Kritharides et al., 1998; Tall & Yvan-Charvet, 2015; Saha et al., 2017). More information about analysis and peak characterization provided in Appendix E.

# 5. Conclusion

In summary, we have presented an autoencoder-based methodology for hyperspectral unmixing in Raman spectroscopy, which we validated on a wide array of synthetic and experimental datasets. Our results demonstrate that autoencoders are adept at handling diverse mixture scenarios and exhibit robustness against data artifacts, offering an effective, versatile and efficient framework for RS unmixing.

# Acknowledgements

D.G. and G.P. are supported by UK Research and Innovation [UKRI Centre for Doctoral Training in AI for Healthcare grant number EP/S023283/1]. A.F.G. acknowledges support from the Schmidt Science Fellows, in partnership with the Rhodes Trust. S.V.P. acknowledges support from the Independent Research Fund Denmark (0170-00011B). R.X. and M.M.S. acknowledge support from the Engineering and Physical Sciences Research Council (EP/P00114/1 and EP/T020792/1). M.M.S. acknowledges support from the Royal Academy of Engineering Chair in Emerging Technologies award (CiET2021\94) and the Bio Innovation Institute. M.B. acknowledges support from the Engineering and Physical Sciences Research Council (EP/N014529/1, funding the EPSRC Centre for Mathematics of Precision Healthcare at Imperial, and EP/T027258/1). The authors thank Dr Akemi Nogiwa Valdez for proofreading and data management support. Figures were assembled in BioRender.

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

## A. Standard methods for hyperspectral unmixing

N-FINDR and VCA are geometric methods based on the concept of a simplex in Euclidean space. N-FINDR exploits the fact that, under the linear mixing model, endmembers represent vertices of a simplex spanning the data, and operates by iteratively finding a set of points (endmembers) that maximizes the volume of the simplex they form. In contrast, VCA finds endmembers by projecting the data onto directions orthogonal to the subspace spanned by previously found endmembers and identifying new endmembers as the farthest points in these directions, effectively constructing a simplex that encompasses all data points. In both methods, the number of endmembers to extract is specified *a priori* by the user. Once endmember signatures $M$ are derived, optimization-based algorithms such as NNLS and FCLS are employed to estimate the fractional abundances $\boldsymbol{\alpha}$ for a given spectrum $\mathbf{x}$ by minimizing the reconstruction error between the observed data and the model $\min_{\boldsymbol{\alpha}} \|M\boldsymbol{\alpha} - \mathbf{x}\|^2$. NNLS imposes the ANC, whereas FCLS imposes both the ANC and ASC.

## B. Autoencoder architectures

**Dense AE.**   This autoencoder employs an encoder comprising 2 fully connected (or dense) layers. The first layer projects spectra of dimension $b$ to hidden features of dimension 128 (Leaky ReLU activation with a slope of 0.02), which the second layer further projects to latent representations of dimension $n$ ($n$ is the number of endmembers to extract). In the *Deep Dense AE* model used in the analysis of the THP-1 cell, we increase the number of hidden layers to five, comprising 512, 256, 128, 64 and 32 neurons, respectively, before the final layer of size $n$.

**Convolutional AE.**   This model extends the *Dense AE* by adding a convolutional block before the dense layers. The convolutional block consists of two layers of 1D convolutions connected in parallel, each comprising 16 filters of size 3 and 16 filters of size 5 (ReLU activation; input padded with zeroes). The outputs from these two layers are concatenated and merged (channel-wise) via a 2-dimensional dense layer to produce representations of dimension $b$, which are then fed to the *Dense* encoder described above.

**Transformer AE.**   In this transformer-based encoder, input spectra are first projected to features of size 32 through a fully connected layer, and then fed to a transformer encoder layer comprising a multi-head attention block with 2 attention heads of size 32 (Vaswani et al., 2017), followed by two fully connected layers expanding the features to size 64 (ReLU activation) and condensing back to 32 (no activation). We apply layer normalization (Ba et al., 2016) and dropout (10%) (Srivastava et al., 2014) after the multi-head attention block and the fully connected layers. The output of the transformer block is then channeled into the last fully connected layer of size $n$.

**Convolutional Transformer AE.**   In this model, the *Transformer AE* architecture is extended with the same convolutional block used in the *Convolutional AE*, here added before the transformer-based encoder block.

**Decoder choice.**   Our linear unmixing decoder architecture consists of a single fully connected layer using the identity activation function without bias. This results in a layer defined by a $b \times m$ weight matrix $W$, where output reconstructions $\widehat{\mathbf{x}}$ become

$$\widehat{\mathbf{x}} = \mathcal{D}_{\mathrm{Lin}}(\mathbf{z}) = W\mathbf{z}. \tag{1}$$

Our bilinear Fan decoder has the same architecture as the linear decoder but also calculates the additional bilinear interaction terms during each forward pass as follows:

$$\widehat{\mathbf{x}} = \mathcal{D}_{\mathrm{Bilin}}(\mathbf{z}) = W\mathbf{z} + \sum_{k=1}^{m} \sum_{\substack{l=1, \\ l \neq k}}^{m} z_k \mathbf{w}_k \odot z_l \mathbf{w}_l, \tag{2}$$

where $z_k, z_l$ are components of $\mathbf{z}$, and $\mathbf{w}_k, \mathbf{w}_l$ are column vectors of $W$.

## C. Benchmark results on synthetic data

### C.1. Generating synthetic Raman mixtures

The synthetic data generation process we adopt is as follows (see Figure 4a for overview).

**Generating endmembers.** For each synthetic dataset, we first generate $n$ endmembers spanning $b$ spectral bands. For the scope of this work, $n = 5$ and $b = 1000$. Each endmember $\mathbf{m}_i \in \mathbb{R}_+^b$ is created by a superposition of a set of $n_{\text{peaks},i}$ Gaussian peaks of different amplitude, width and location, randomly sampled as follows. The number of peaks is sampled from a discrete uniform distribution $n_{\text{peaks},i} \sim \mathcal{U}(5, 9)$. Each peak $p$ is described by $p = h_p \sigma_p \sqrt{2\pi} \, \mathcal{N}(b_p, \sigma_p)$, where $\mathcal{N}(\cdot)$ represents a Gaussian distribution. The height of the peak is defined as $h_p = h_1 \cdot h_2$, where $h_1 = 1 + 5 \, h_\beta$ with $h_\beta \sim \text{Beta}(1, 3)$ and $h_2 \sim \mathcal{U}(0.1, 1)$. The center of the peak is sampled from $b_p \sim \mathcal{U}(10, b - 10)$, and the width of the peak is defined as $\sigma_p = w_p \sigma$, with $\sigma \sim \mathcal{U}(0.1, 1)$. We create two types of endmembers: *clean* and *noisy*. For the former, we produce peaks with $w_p = 1$. For the latter, we augment *clean* endmembers by adding $n_{\text{peaks},i}^{\text{small}} \sim \mathcal{U}(50, 99)$ smaller peaks sampled with $h_1 = 1/3$ and $w_p = 2$, thus making *noisy* endmembers better resemble experimental Raman signatures.

**Generating fractional abundances.** For visualization purposes, we present the fractional abundance profiles in the form of two-dimensional scenes comprising $H \times W$ pixels, where each pixel represents a fractional abundance vector $\boldsymbol{\alpha} \in \mathbb{R}_+^n$. Here, we set $H = W = 100$, resulting in $10000$ spectra per scene/dataset. In the simplest scene (*Chessboard*), we split the scene into $20 \times 20$ square patches, each containing a single randomly assigned endmember (i.e., all $400$ pixels in each patch are the same one-hot vector). Our second scene (*Gaussian*) consists of $n$ Gaussian functions equally spaced along the diagonal of the scene. After each pixel is normalized to comply with the ASC, we obtain abundance profiles representing different levels of overlap of components. Our last fractional abundance scene (*Dirichlet*) corresponds to a highly mixed scene, where each pixel is individually sampled from a $n$-dimensional Dirichlet distribution, producing a random mixture of all endmembers. Note that the fractional abundance profile of each pixel in all three scenes complies with both ANC and ASC.

**Mixing model.** Having generated a set of endmembers and an underlying fractional abundance scene, mixed data measurements $\mathbf{x} \in \mathbb{R}^b$ are created based on a mixing model chosen by the user. In this study, we consider linear mixtures and bilinear mixtures based on the Fan model.

**Adding data artifacts.** Finally, data artifacts (noise, baseline, cosmic spikes) can be optionally added to create more realistic synthetic Raman spectra. Here, we add Gaussian noise $\boldsymbol{\epsilon} \in \mathbb{R}^b$ to each spectrum, with independent and identically distributed components $\epsilon_i \sim \mathcal{N}(0, \sigma_N)$. Further, we add a baseline signal $\mathbf{B} = h_B \arctan(\pi[1 : b]/b) \in \mathbb{R}^b$ to each spectrum with probability $p_B$. Finally, with probability $p_S$, a cosmic spike of intensity $S \sim h_S U(0.75, 1.25)$ is added to each spectrum at a band $b_S \sim \mathcal{U}\{2, b - 2\}$. In our experiments: $\sigma_N = 0.1, p_B = 0.25 \, h_B = 2, , p_S = 0.1, h_S = 5$.

## C.2. Synthetic datasets

Using the generator we developed, we can produce synthetic Raman mixtures with different characteristics (e.g., number and type of endmembers, abundance profiles, mixture model, data artifacts) with full knowledge of the 'ground truth' endmembers and fractional abundances. This allows us to quantify and compare the performance of unmixing approaches (see Figure 4b for unmixing of an example synthetic dataset).

Using our data generator, we produce 11 types of synthetic datasets of variable complexity, based on four mixture scenarios over three fractional abundance scenes. In order of complexity, the four mixture scenarios are: 1) a linear mixture with *clean* endmembers and no data artifacts (*ideal*); 2) a linear mixture with *clean* endmembers, but contaminated with artifacts representing dark noise, baseline variations and cosmic spikes (*+artifacts*); 3) a linear mixture with *noisy* endmembers (i.e., containing additional smaller noise peaks) and artifacts (*+realistic*); and 4) a bilinear mixture based on the Fan model with *noisy* endmembers and artifacts (*+bilinear*). For each of the four mixture scenarios, we generate three dataset variants (two for the *+bilinear* scenario since no bilinear interactions are present in our *Chessboard* scene) based on custom $100 \times 100$ fractional abundance scenes. This produces 10k spectra per dataset, organized into two-dimensional scenes for visualization purposes. In increasing level of mixing, we have: 1) a scene comprising well-separated patches, each containing a single species (*Chessboard* scene); 2) a semi-mixed scene given by a Gaussian mixture of species (*Gaussian* scene); and 3) a highly-mixed scene where each pixel represents a random sample of species drawn from a Dirichlet distribution (*Dirichlet* scene). Thus, our synthetic datasets cover varied mixing scenarios, from the *ideal Chessboard* dataset, which is trivial for conventional methods, to noisier, more complex mixtures containing different types of artifacts.

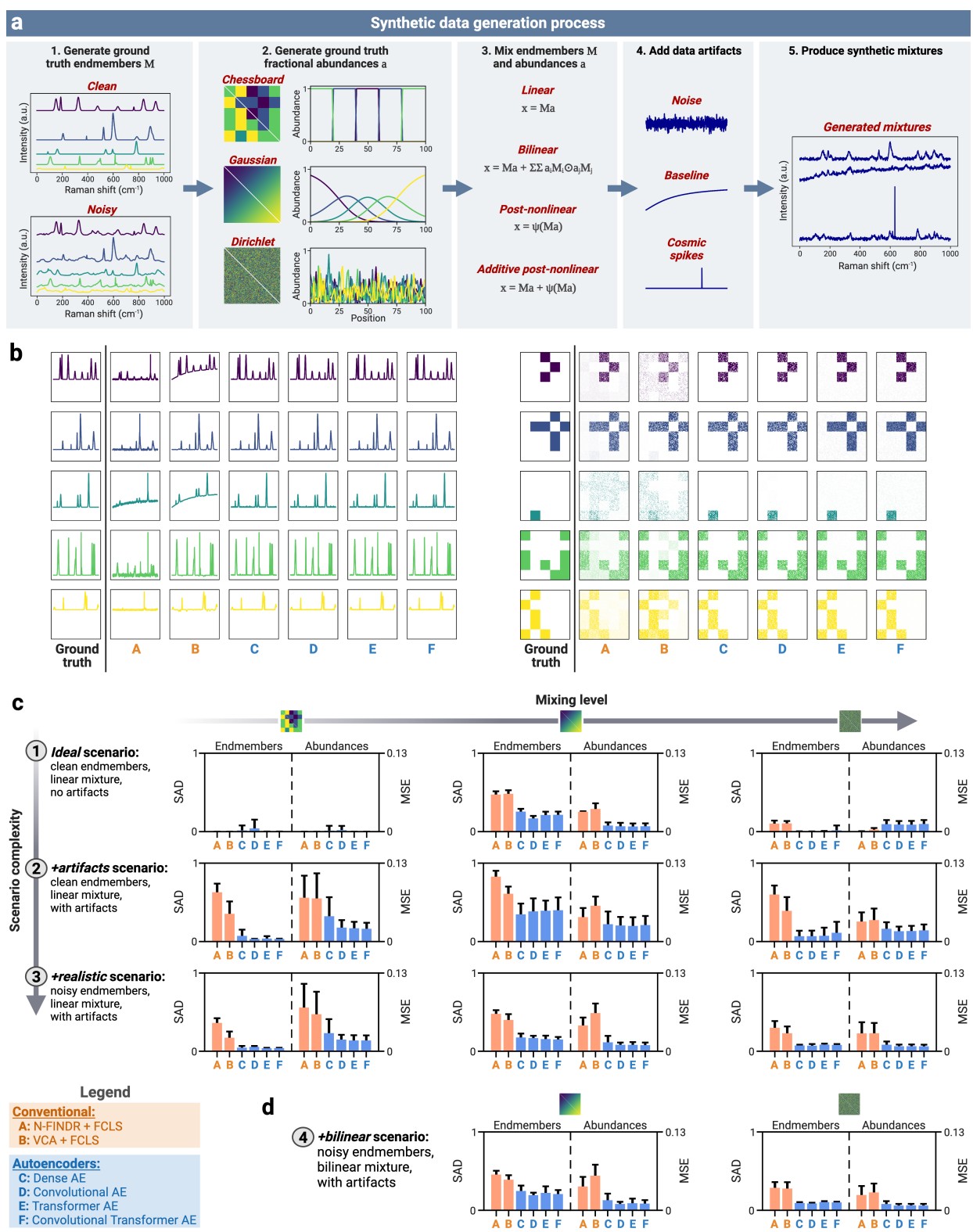

*Figure 4.* Benchmarking autoencoders on synthetic Raman mixtures. a, Schematic of our synthetic data generation workflow. b, Representative results for the six algorithms (two conventional and four AEs) on an example synthetic dataset (*Chessboard+artifacts* scenario): endmembers (*left*), and fractional abundances (*right*). c-d, Summary of unmixing performance on synthetic datasets of variable mixing level and complexity: linear mixtures (c), bilinear mixtures (d). Confidence intervals are given by one standard deviation around the sample mean ($n = 25$ samples: 5 datasets with 5 model repetitions each).

## C.3. Model training and evaluation on synthetic Raman mixtures

Autoencoders were trained on synthetic data using the Adam optimizer (learning rate 0.001) over 10 epochs, with spectral angle distance as a loss function between input and reconstructed spectra. The latent dimensionality $m$ of each AE model is set to 5 for the *ideal* mixture scenario, and 6 for the other mixture scenarios with data artifacts. Both ANC and ASC are enforced for all experiments on synthetic data. Each experiment on the synthetic data was performed on 5 datasets and 5 model initializations using different random seeds, resulting in $5 \times 5 = 25$ replicates per evaluation, or 1650 experiments in total: 6 models (2 conventional, 4 AEs) $\times$ 11 dataset variants $\times$ 25 replicates. Random seeds were kept the same across mixture scenarios to allow direct comparison.

## C.4. Benchmark results

**Benchmark results on linear mixtures.**  We first discuss our results on the nine dataset variants created through the linear mixture scenarios (1-3). Such data complies with the linear mixing assumption of conventional methods and, for consistency, we equip the AE models with a decoder for linear unmixing. Figure 4c summarises the performance of the six models (two conventional and four AEs) across the nine dataset variants, with experiments performed over 5 distinct datasets and 5 model initializations for each variant. We find that the AE models outperform the two conventional methods, providing more accurate endmembers and fractional abundances across virtually all scenarios and abundance scenes. The AEs recover the performance of the conventional methods on the simple *ideal Chessboard* datasets, and the improvement in AE performance becomes increasingly prominent for mixture scenarios with higher levels of noise and data artifacts.

**Non-linear unmixing with autoencoders.**  We next proceed to our benchmark analysis on synthetic data generated using a non-linear mixture model (i.e., +*bilinear* scenario). This time, we equip AEs with a decoder specific to the bilinear mixture model, which is achieved by merely adapting the decoder architecture. Our experimental results are displayed in Figure 4d. Again, we observe that all four AE models provide a substantial improvement in unmixing accuracy compared to standard unmixing methods for both endmember and abundance estimation.

## C.5. Computational efficiency

The computational complexity and scalability of unmixing methods can become a significant bottleneck in real-world applications, particularly for imaging and volumetric Raman scans, which can contain hundreds of thousands of spectra. To examine this issue, we profile the computational cost of our four AE methods (linear decoders) and the two conventional methods on synthetic datasets (*ideal* scenario, *Chessboard* scene) of increasing size up to 250000 spectra. The number of endmembers to extract was set to $n = 5$ for all methods. For each experiment, we performed 3 separate evaluations, measuring the wall time of each method (including the training time for autoencoders). All experiments were conducted on a MacBook Air laptop with an Apple M2 chip (8-core CPU, 10-core GPU, and 16-core Neural Engine). To be fair to conventional algorithms, we include the full training time for autoencoders and use CPU computation to avoid any advantage from GPU acceleration.

Figure 5 shows that all AE models are faster than N-FINDR+FCLS and VCA+FCLS, which are already among the most computationally lightweight conventional unmixing techniques (Bioucas-Dias et al., 2012).

# D. Analysis of experimental RS data from sugar mixtures

## D.1. Preparation of sugar solutions

We prepared $1\,\mathrm{mol/L}$ solutions of each type of sugar (sucrose, fructose, maltose, and glucose) by dissolving the appropriate weight of sugar into $40\,\mathrm{mL}$ of ultrapure distilled water (Invitrogen™ – UltraPure™ DNase/RNase-Free Distilled Water). The weights of sugars dissolved were $13.83\,\mathrm{g}$ for sucrose (Thermo Scientific Chemicals – Sucrose, 99%), $7.279\,\mathrm{g}$ for fructose (Thermo Scientific Chemicals – D-Fructose, 99%), $15.171\,\mathrm{g}$ for maltose (Thermo Scientific Chemicals – D-(+)-Maltose monohydrate, 95%) and $7.279\,\mathrm{g}$ for glucose (D-(+)-Glucose, AnalaR NORMAPUR® analytical reagent). All solutions were mixed and vortexed in standard $50\,\mathrm{mL}$ centrifuge tubes until no solute was visible.

Sugar mixtures were prepared in standard 96-well plates, with a volume of $375\,\mathrm{\mu L}$ per well. A full factorial experiment was performed comprising 4 volume levels for each sugar ($0\,\mathrm{\mu L}$, $30\,\mathrm{\mu L}$, $75\,\mathrm{\mu L}$ and $120\,\mathrm{\mu L}$), filled with distilled water where necessary. Discarding the mixtures exceeding the volume of the well and the one that contains no sugar, 240 distinct

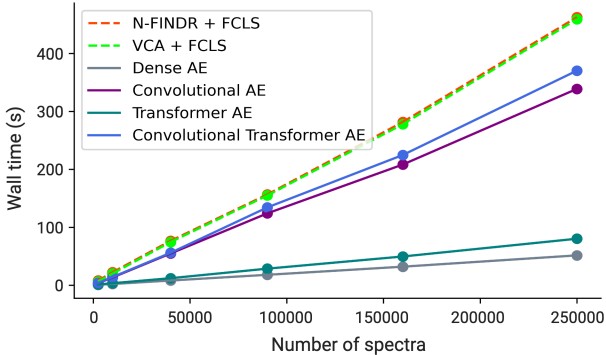

*Figure 5.* Computational efficiency of autoencoders and conventional methods on synthetic datasets with an increasing number of spectra. Each dot represents the average across 3 evaluations (confidence intervals based on one standard deviation are small and not visible to the eye). AE models are equipped with decoders for linear unmixing. Data generated under *Chessboard +artifacts*.

mixtures were prepared. In addition, 5 extra 'pure' solutions (i.e., $375\,\mu L$ of water, sucrose, fructose, maltose, or glucose) were prepared, which we used to extract reference spectra for each chemical species. This resulted in a total of 245 wells distributed in three standard 96-well plates. Mixtures were stirred using standard $200\,\mu L$ pipettes before spectral acquisition to ensure good mixing.

### D.2. Raman measurements from sugar solutions

All spectra were acquired using a custom Raman microspectroscopy platform designed for high-throughput analysis known as B-Raman. This platform is based on the Thorlabs Cerna® and features the BWTek BRM-785-0.55-100-0.22-SMA laser excitation source and the Ibsen EAGLE Raman-S spectrometer. The instrument was calibrated using an Argon wavelength calibration source (AR-2 – Ocean Insight) reference lamp before data collection. The excitation wavelength was $785\,nm$ and the power incident to the samples was $36.3\,mW$. The Raman scattering was collected in reflection via a Leica N PLAN 10x/0.25 objective with $0.25$ numerical aperture. The raw spectra were acquired over the spectral wavenumber range of $142–3684.8\,cm^{-1}$.

Spectra were measured from the center (horizontal) of each well at a fixed depth that was established to provide the highest signal. Two sets of data were collected from each well, at $5\,s$ and $0.5\,s$ integration times, to compare unmixing performance on low and high signal-to-noise ratio (SNR) data. Several measurements were collected from each well, resulting in a total of 240 solutions $\times$ 2 measurements $\times$ 4 repetitions $=$ 1920 high-SNR measurements (1960 with reference spectra); and 240 solutions $\times$ 8 measurements $\times$ 4 repetitions $=$ 7680 low-SNR measurements (7840 with reference spectra). Ground-truth endmembers signatures were obtained by taking the median (band-wise) of the reference spectra (40 in high SNR setup, and 160 in low SNR setup) collected from the 5 additional wells containing pure solutions. Ground truth fractional abundances were determined by calculating the ratio of the components present in each mixture.

### D.3. Preprocessing and analysis of sugar data

First, we preprocess each sugar dataset: 1) cropping to the region $400–1800\,cm^{-1}$; 2) baseline correction with Adaptive Smoothness Parameter Penalized Least Squares (ASPLS) (Zhang et al., 2020a)—smoothing parameter $\lambda = 10^5$, differential matrix of order 2, maximum iterations set to 100, exit criteria with tolerance $t = 0.001$; 3) global vector normalization, where each observation is divided by the highest magnitude observed in the data. Baseline removal is important to ensure models extract relevant features (i.e., characteristic peaks) as opposed to merely capturing the trend.

To perform hyperspectral unmixing, we set the number of endmembers to extract to $n = 5$, and we follow similar training and evaluation protocols to those employed for the synthetic data, but we increase the number of epochs to 15 for low SNR data and 50 for high SNR data given the more limited number of spectra collected. We also incorporate an additional MSE term in the training loss $\mathcal{L}$ of autoencoders on high SNR data:

$$\mathcal{L}(\mathbf{x}, \widehat{\mathbf{x}}) = \text{SAD}(\mathbf{x}, \widehat{\mathbf{x}}) + \lambda\,\text{MSE}(\mathbf{x}, \widehat{\mathbf{x}}), \tag{3}$$

with $\lambda = 1000$. This term breaks the invariance to scale and leads to better abundance estimation given the weak water

endmember. The standard SAD loss was used for experiments on low SNR. Each experiment is repeated for 5 model initializations.

## E. Analysis of volumetric RS data from THP-1 cell

The volumetric Raman scan of the THP-1 cell (Kallepitis et al., 2017) was collected using $0.3\,$s integration time and comprises a $z = 1, \ldots, 10$ stack of ten $40 \times 40$ raster scans, organized into a single volumetric hypercube for analysis. We preprocess the data before unmixing using the following protocol: 1) spectral cropping to the fingerprint region 700–$1800\,$cm$^{-1}$; 2) cosmic spike removal using the algorithm in (Whitaker & Hayes, 2018) with kernel of size 3 and z-value threshold of 8; 3) denoising with Savitzky-Golay filter using a cubic polynomial kernel of size 7 (Savitzky & Golay, 1964); 4) baseline correction using Asymmetric Least Squares (AsLS) with smoothing parameter $\lambda = 10^6$, penalizing weighting factor $p = 0.01$, differential matrix of order 2, maximum iterations set to 50, exit criteria with tolerance threshold of $t = 0.001$ (Eilers & Boelens, 2005); 5) global MinMax normalization to the interval $[0, 1]$.

Unmixing is performed following the same AE training protocol as in other analyses, with the number of training epochs set to 20, and the number of endmembers to extract to $n = 20$. Here, we also discard the constraint that fractional abundances must sum to one. Out of the 20 endmembers we obtain, we display the 5 deemed most biologically relevant following peak assignment as per the original paper (Kallepitis et al., 2017). For VCA+NNLS, two of those five endmembers corresponded to the same cell organelle, namely cytoplasm, and were visualized using the same color in the merged reconstruction displayed in Fig. 3b.

Cell organelles were determined based on the following peaks: PBS buffer - $1637\,$cm$^{-1}$ (water peak); cytoplasm - $1005\,$cm$^{-1}$ (phenylalanine), $1250\,$cm$^{-1}$ (Amide III), $1659\,$cm$^{-1}$ (Amide I) and $1445\,$cm$^{-1}$ (CH deformations of proteins and lipids); TAGs/PLPs - $1092\,$cm$^{-1}$ (C–C stretching ), $1308\,$cm$^{-1}$ (CH$_2$ twists), $1445\,$cm$^{-1}$ (CH deformation), and $1661\,$cm$^{-1}$ (C=C stretching); nucleus/DNA - $790\,$cm$^{-1}$ (symmetric phosphodiester stretch and ring breathing modes of pyrimidine bases) and $1103\,$cm$^{-1}$ (symmetric dioxy-stretch of the phosphate backbone); cholesterol - $1069\,$cm$^{-1}$ and $1134\,$cm$^{-1}$ (cholesteryl stearate), $1300\,$cm$^{-1}$ (CH$_2$ twists), and $1443\,$cm$^{-1}$ (CH deformation) (Kallepitis et al., 2017; Zhang et al., 2012; Movasaghi et al., 2007).