# OpenReview forum: "Hyperspectral Unmixing for Raman Spectroscopy via Physics-Constrained Autoencoders"
_ICML.cc/2024/Workshop/ML4LMS — ML4LMS Poster_

### Official Review · Reviewer_4yjy · 2024-06-03
**Very good, translational paper**

**Rating:** 9
**Confidence:** 3

**Review:**

The authors propose a novel use of autoencoders to extract the pure, original components of an observed Raman spectra. They validate their models with synthetic and experimental data, obtaining very compelling results. Overall, this work displays a very large degree of applicability in the life sciences, showcased by their experiment involving living cells, being able to identify a variety of components present in them. I propose the acceptance and the consideration of this paper for the life sciences award.

---

### Official Review · Reviewer_RQwB · 2024-06-12
**Autoencoders for analysis of Raman Spectroscopy Data**

**Rating:** 6
**Confidence:** 3

**Review:**

The paper is well written and the results are interesting. Just submitting the score for now, I will aggregate my comments on the paper and edit the response in the next couple of days

---

### Official Review · Reviewer_Nbpw · 2024-06-12
**Hyperspectral Unmixing for Raman Spectroscopy**

**Rating:** 6
**Confidence:** 2

**Review:**

The paper was clear and quite well written. I cannot speak much to the originality or significance.

Pros:
-good description of methods
-precise definition of task
-enough info for reproduction if wanted
-baselines included

Cons:
-unclear about what the baselines are
-lack of training and validation results (it would be nice to see a quantitative analysis on how well these methods perform against eachother)
- does not seem overly significant as it relies on a very specific spectral method